# CD200R deletion promotes a neutrophil niche for *Francisella tularensis* and increases infectious burden and mortality

J. Casulli[1,2,3], M.E. Fife[1,3], S.A. Houston[1,2,3], S. Rossi[1,2,3], J. Dow[1,2,3], E.D. Williamson[4], G. C Clark[4], T. Hussell[1,3], R.V. D'Elia[4] & M. A Travis[1,2,3]

Pulmonary immune control is crucial for protection against pathogens. Here we identify a pathway that promotes host responses during pulmonary bacterial infection; the expression of CD200 receptor (CD200R), which is known to dampen pulmonary immune responses, promotes effective clearance of the lethal intracellular bacterium *Francisella tularensis*. We show that depletion of CD200R in mice increases in vitro and in vivo infectious burden. In vivo, CD200R deficiency leads to enhanced bacterial burden in neutrophils, suggesting CD200R normally limits the neutrophil niche for infection. Indeed, depletion of this neutrophil niche in CD200R$^{-/-}$ mice restores *F. tularensis* infection to levels seen in wild-type mice. Mechanistically, CD200R-deficient neutrophils display significantly reduced reactive oxygen species production (ROS), suggesting that CD200R-mediated ROS production in neutrophils is necessary for limiting *F. tularensis* colonisation and proliferation. Overall, our data show that CD200R promotes the antimicrobial properties of neutrophils and may represent a novel antibacterial therapeutic target.

[1] Lydia Becker Institute for Immunology and Inflammation, Manchester, UK. [2] Wellcome Trust Centre for Cell-Matrix Research, Manchester, UK. [3] Manchester Collaborative Centre for Inflammation Research (MCCIR), Faculty of Biology, Medicine and Health, Manchester Academic Health Sciences Centre, University of Manchester, Manchester, UK. [4] Defence Science and Technology Laboratory (Dstl), Porton Down, Salisbury, UK. Correspondence and requests for materials should be addressed to M.A.T. (email: mark.travis@manchester.ac.uk)

*F*rancisella tularensis is a highly infectious Gram-negative intracellular bacterium and is the causative agent of the lethal disease tularaemia[1]. *F. tularensis* can cause infection by multiple routes; however, infection via the respiratory route can lead to the most severe form of the disease, with as little as 25 colony-forming units (CFUs) causing life-threatening infection if left untreated[2–4]. Due to its high virulence via the aerosol route, *F. tularensis* has long been considered a potential biological weapon[5,6]. However, at present there is no licensed vaccine available against *F. tularensis*[7–9] and much is still unknown about how the bacterium causes lethal infection.

Murine models of infection provide evidence that *F. tularensis* can initially evade the host innate immune response[10,11]. Production of pro-inflammatory cytokines is delayed up to 72 h post infection (p.i.), which is followed by a rapid cytokine response that results in sepsis and death of the host[12–14]. However, the mechanisms that regulate development of a protective immune response during *F. tularensis* infection are unclear. Thus, there is an urgent need to understand better how the host responds to infection with *F. tularensis*, in order to identify potential novel targets to promote beneficial immune responses.

*F. tularensis* infects various types of immune cells in the lung, predominantly macrophages, during early respiratory infection and neutrophils from day 3 p.i., using them as a replicative niche[15,16]. We therefore hypothesised that expression of regulatory molecules by innate cells such as macrophages and neutrophils may limit immune responses to *F. tularensis* infection.

CD200 receptor (CD200R) is a regulatory receptor prominently expressed in the lungs, with expression observed on alveolar macrophages and neutrophils, and its ligand CD200 expressed primarily by alveolar epithelial cells[17,18]. Therapeutically targeting the CD200R pathway is beneficial in alleviating influenza-induced inflammation, reducing severity of arthritis and modulating microglial activation in neurodegenerative disease[19–21]. Therefore, we hypothesised that CD200R may play a critical role in suppression of immune responses to *F. tularensis*, and that targeting CD200R could improve protective immunity against *F. tularensis*.

To explore this hypothesis, we used models of infection using *F. tularensis* live vaccine strain (LVS), which causes a lethal infection in mice[7]. We demonstrate that a lack of CD200R, instead of boosting immunity to *F. tularensis*, surprisingly led to enhanced bacterial burden in vitro and in vivo. Indeed, stimulation of the CD200R pathway in vivo by its ligand CD200 promoted enhanced clearance of bacterial burden. Exacerbated bacterial burden in CD200R$^{-/-}$ mice was due to increased neutrophil influx, with neutrophils in CD200R$^{-/-}$ mice showing enhanced levels of infection. Depletion of neutrophils in CD200R$^{-/-}$ mice rescued infectious burden to levels observed in wild-type (WT) mice, strongly indicating that the CD200R$^{-/-}$ neutrophils are acting as a niche for enhanced infectious burden. Mechanistically, we show that CD200R-deficient neutrophils display dysfunctional reactive oxygen species (ROS) production, suggesting that the CD200R pathway has a key role in the antimicrobial responses in neutrophils. Thus, our data highlight a previously unappreciated role for the CD200R pathway in limiting bacterial infection in neutrophils, which would be amenable to therapeutic promotion in vivo.

## Results

### Lack of the CD200R pathway promotes *F. tularensis* infection.
To explore the role for CD200R during *F. tularensis* infection, we first infected bone marrow-derived macrophages (BMDMs) or neutrophils, cells previously shown to be key targets for *F. tularensis* infection[15]. Cells were isolated from WT or CD200R$^{-/-}$

mice, and infected with *F. tularensis*. We observed a significantly enhanced bacterial burden via flow cytometry in CD200R$^{-/-}$ BMDM at 2 h p.i. compared with WT cells, with the increased burden becoming even more significant at 24 h p.i. (Fig. 1a, b). There was also a significantly enhanced bacterial burden in CD200R$^{-/-}$ neutrophils at 24 h p.i. (Fig. 1c, d). These differences were also observed when bacterial burden was assessed via bacteriology (Supplementary Fig. 1A, B). Differences do not appear to be influenced by alterations in cell mortality, as both WT and CD200R$^{-/-}$ BMDMs and neutrophils show similar levels of cell death in assays (Supplementary Fig. 1C, D).

As CD200R requires interaction with its ligand CD200 to signal, we next assessed CD200 levels on bone marrow macrophages and neutrophils by flow cytometry. We found that there was low but detectable expression of CD200 ligand on the surface of both cell types (Supplementary Fig. 2A–F). There appears to be enhanced levels of CD200 on the surface of CD200R$^{-/-}$ macrophages (Supplementary Fig. 2A–C), but not neutrophils (Supplementary Fig. 2D–F). Therefore, there is potential for CD200 expression by both macrophages and neutrophils to stimulate CD200R in an autocrine fashion in our assays, with this stimulation not possible in CD200R-deficient cells. However, we cannot rule out contribution of other CD200 ligand cellular sources in vivo, playing a role in signalling to cells prior to their isolation and use in assays.

Together, these data suggest that CD200R, instead of inhibiting responses to infection as predicted given its known regulatory function, plays an important role in controlling *F. tularensis* infectious outcome in vitro.

### CD200R$^{-/-}$ mice display enhanced *F. tularensis* burden.
Having confirmed that lack of CD200R enhances *F. tularensis* infection of macrophages and neutrophils in vitro, we next assessed whether mice lacking CD200R were more susceptible to infection. WT and CD200R$^{-/-}$ mice were infected via the intranasal route with 1000 CFU *F. tularensis* and monitored for 7 days. CD200R$^{-/-}$ mice displayed significantly decreased survival at day 7 p.i. with *F. tularensis* compared with WT mice (Fig. 2a). Bacterial burden in the lung showed no significant differences at early time points p.i. (day 1, 3 or 5 p.i.) (Fig. 2b). However, we saw a significantly enhanced bacterial burden at day 7 p.i. in CD200R$^{-/-}$ mice (Fig. 2b). This was also accompanied by significantly increased splenomegaly in CD200R$^{-/-}$ mice compared with WT at day 7 p.i. (Fig. 2c). These data show that lack of CD200R expression in mice results in exacerbated bacterial burden and infectious outcome following *F. tularensis* infection.

*F. tularensis* predominantly infects pulmonary macrophages and neutrophils[15]. Given that both these types of cell showed enhanced infectious burden in the absence of CD200R expression in vitro, we hypothesised that there may be increased infection of macrophages and neutrophils in vivo in CD200R$^{-/-}$ mice. To test this hypothesis, we analysed infection in both cell types via flow cytometry, identifying neutrophils as CD45$^+$CD11b$^+$ Ly6G$^+$ and macrophages as CD45$^+$Ly6G$^-$SiglecF$^-$CD64$^+$ CD11b$^+$CD11c$^{INT}$;[22,23] Supplementary Fig. 3) Interestingly, we found that the increased bacterial burden in CD200R$^{-/-}$ mice was restricted to neutrophils, alongside a dramatic increase in neutrophils (both as a % and absolute number) in the infected CD200R$^{-/-}$ lung at days 5 and 7 p.i. (Fig. 2d, e and Supplementary Fig. 4A, B). In contrast, macrophages did not exhibit any changes in either bacterial burden or in infiltration into the lung, apart from a slight decrease in percentage of macrophages at day 5 p.i. (Fig. 2f, g and Supplementary Fig. 4C, D). The increased neutrophil numbers in CD2000R$^{-/-}$ mice did

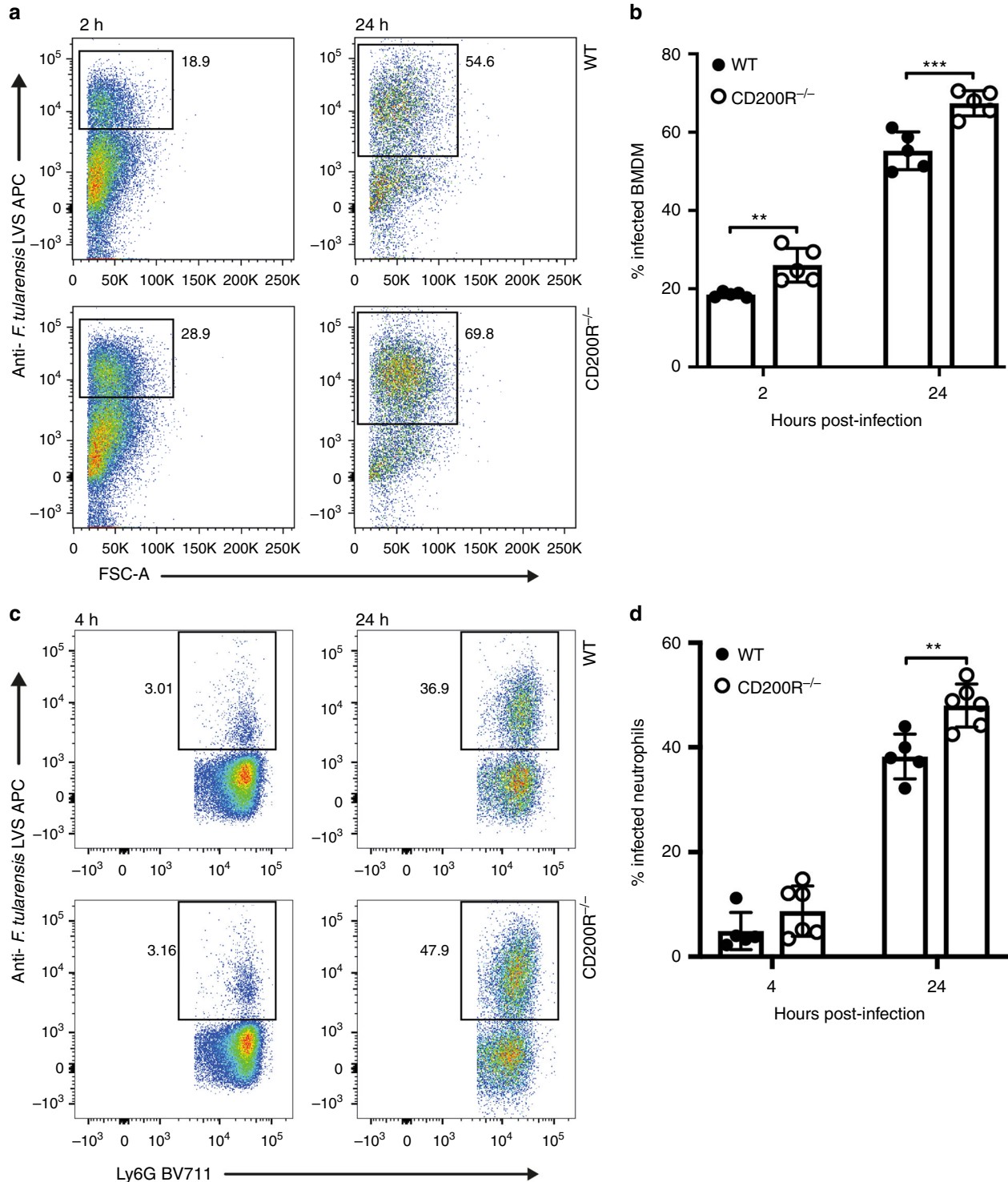

**Fig. 1** Macrophages and neutrophils lacking CD200R display enhanced *F. tularensis* burden. Primary BMDM or bone marrow neutrophils were derived from wild-type or CD200R$^{-/-}$ mice and infected with *F. tularensis* LVS MOI 100. **a** Representative dot plots of *F. tularensis*-infected BMDM at 2 and 24 h p.i. Cells were labelled with an anti-*F. tularensis* LVS antibody and samples were pre-gated on live, single cells. **b** Quantification of *F. tularensis* levels in BMDM at 2 and 24 h p.i. **c** Representative dot plots of *F. tularensis*-infected neutrophils at 4 and 24 h p.i. Samples were pre-gated on live, single, CD11b$^+$Ly6G$^+$ cells. **d** Quantification of *F. tularensis* levels in neutrophils at 4 and 24 h p.i. Data represent two independent experiments and is shown as mean ± SD (*n* = 5–6). Statistical analysis was performed using two-way ANOVA (*$p < 0.05$, **$p < 0.01$, ***$p < 0.001$)

not appear to be due to enhanced survival of these cells, as there was a slight decrease in the percentage of live neutrophils observed at day 5 p.i., (Supplementary Fig. 4E) and no difference at day 7 p.i. (Supplementary Fig. 4F). Collectively, these data suggest that the increased burden in the lung of CD200R$^{-/-}$ mice

is predominantly due to increased infection of infiltrating neutrophils that act as a niche for infection.

As lack of CD200R expression resulted in enhanced bacterial burden after *F. tularensis* infection, we next tested whether stimulating this pathway could promote bacterial clearance. To

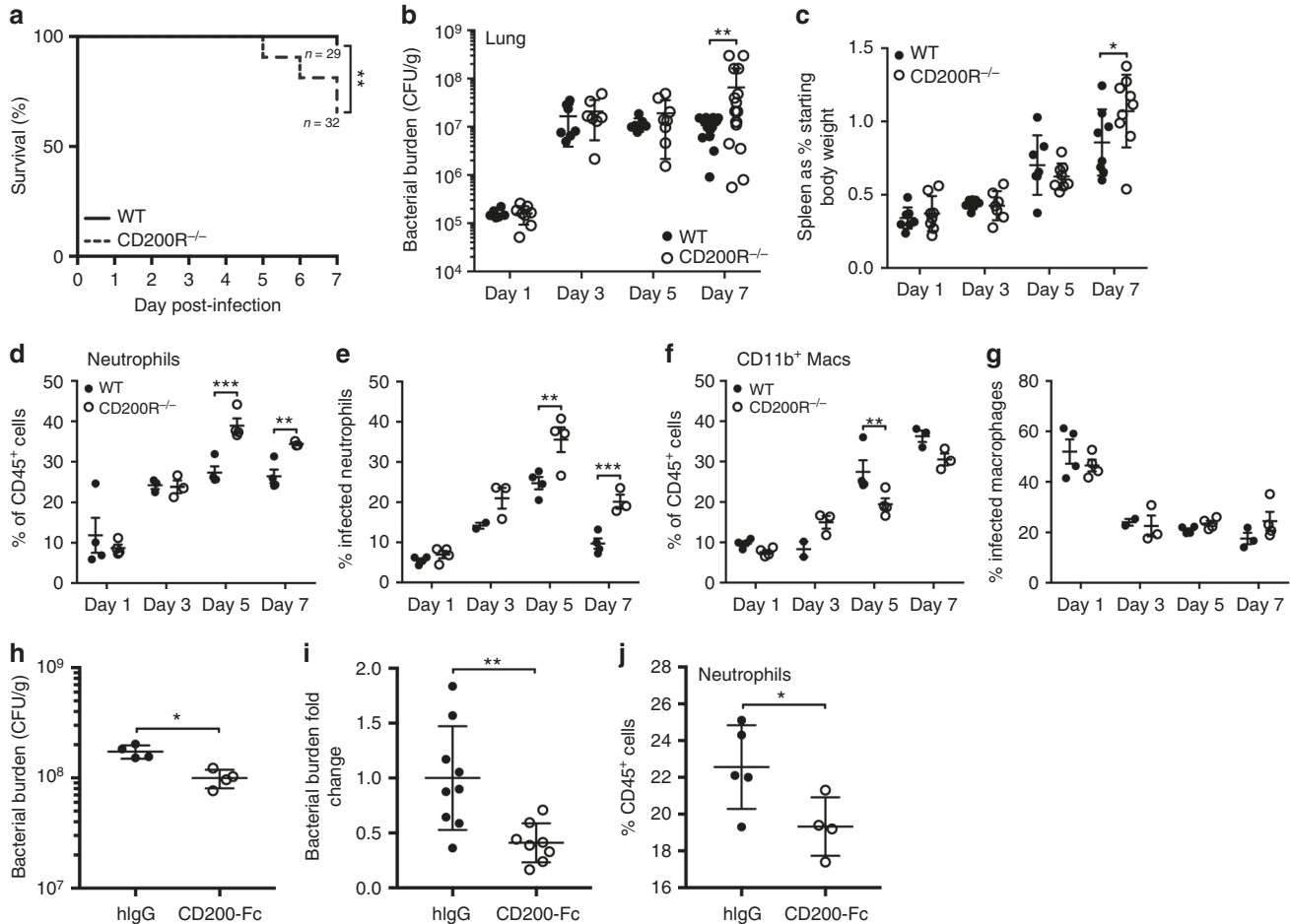

**Fig. 2** Lack of CD200R in mice results in enhanced pulmonary *F. tularensis* burden associated with increased neutrophil influx, displaying exacerbated infectious burden. WT and CD200R$^{-/-}$ C57BL/6 mice were infected IN with a challenge dose of 1000 CFU *F. tularensis* LVS. **a** Percentage survival of WT and CD200R$^{-/-}$ mice over 7 days of infection, with lack of survival considered if mice reached the humane endpoint (see methods). Data represent five independent experiments (WT $n = 29$, CD200R$^{-/-}$ $n = 32$). **b** Bacterial burden (CFU/g) of the lung was enumerated at day 1, 3, 5 and 7 p.i. **c** Spleen size at day 1, 3, 5 and 7 p.i. presented as a percentage of body weight at day 0. **d** Quantification of neutrophils by flow cytometry as a percentage of total CD45 + cells at day 1, 3, 5 and 7 p.i. **e** Percentage of neutrophils infected with *F. tularensis* at day 1, 3, 5 and 7 p.i. determined by flow cytometry. **f** Quantification of CD11b$^+$ macrophages by flow cytometry as a percentage of total CD45$^+$ cells at day 1, 3, 5 and 7 p.i. **g** Percentage of CD11b$^+$ macrophages infected with *F. tularensis* at day 1, 3, 5 and 7 p.i. determined by flow cytometry. **h–j** WT mice were infected IN with 1000 CFU *F. tularensis* and treated with 10 µg CD200-Fc or hIgG IN at day 3 p.i. **h** Bacterial burden (CFU/g) of the lung was enumerated at day 7 p.i. **i** Bacterial burden fold change in lung of CD200-Fc-treated mice against hIgG-control-treated mice. **j** Quantification of neutrophils as a percentage of total CD45+ cells at day 7 p.i. Data represent two to four independent experiments (**b**, **c**, **i**: $n = 7$–19), are representative of two independent experiments (**d–h**, **j**: $n = 3$-4 for each experiment) and is shown as mean ± SD. Statistical analysis was performed using Mantel–Cox test (**a**), two-way ANOVA (**b–g**) and unpaired *t*-tests (**h–j**) (*$p < 0.05$, **$p < 0.01$, ***$p < 0.001$)

this end, we treated mice with CD200 ligand (CD200-Fc)[24] 3 days p.i., to better mimic how treatment would be given therapeutically. We found that treatment of mice caused a significant reduction in bacterial burden (Fig. 2h), with a >50% reduction in bacterial burden achieved (Fig. 2i). In addition, we observed decreased neutrophils in the lung p.i. following CD200-Fc treatment (Fig. 2j). Thus, these data show that targeting the CD200-CD200R pathway p.i. can reduce overall bacterial burden, potentially as a consequence of reduced neutrophil influx, and is therefore a potential therapeutic target for promoting beneficial responses after *F. tularensis* infection.

**Depletion of neutrophils decreases *F. tularensis* burden in CD200R$^{-/-}$ mice.** Next, we aimed to determine important mechanisms by which lack of CD200R promoted bacterial burden post-*F. tularensis* infection. As neutrophils appear to

provide an enhanced niche for *F. tularensis* infection in the absence of CD200R expression, we hypothesised that depletion of this enhanced infectious niche would rescue the exacerbated bacterial burden observed in CD200R$^{-/-}$ mice. To directly test this hypothesis, we treated WT and CD200R$^{-/-}$ mice with an anti-Ly6G monoclonal antibody (clone 1A8) to specifically deplete neutrophils during *F. tularensis* infection. Anti-Ly6G antibody treatment resulted in a ~90% depletion of neutrophils (Fig. 3a, b), with a similar depletion observed in WT and CD200R$^{-/-}$ mice (Fig. 3b). Depletion of neutrophils in CD200R$^{-/-}$ mice rescued the bacterial burdens so that it was comparable to those in WT mice (Fig. 3c). Thus, depletion of neutrophils prevented the enhanced bacterial burden observed in CD200R$^{-/-}$ mice. These data suggest that the higher neutrophil numbers seen in the absence of CD200R signalling provide a replicative niche, aiding in the development of an exacerbated *F. tularensis* infection.

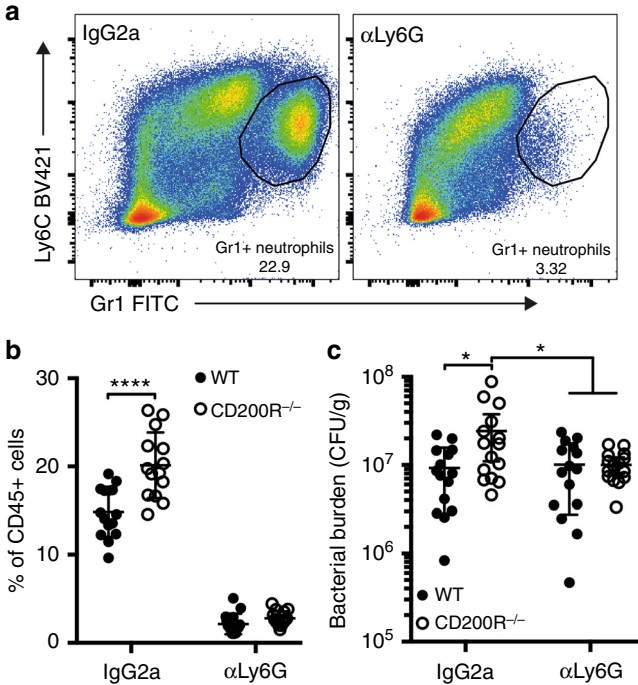

**Fig. 3** Neutrophil depletion rescues the enhanced bacterial burden in the lungs of CD200R$^{-/-}$ mice. WT and CD200R$^{-/-}$ C57BL/6 mice were treated IP with 50 µg anti-Ly6G or Rat IgG2a at day −1, 1, 3 and 5 of *F. tularensis* infection. The challenge dose of ≈1000 CFU *F. tularensis* LVS was administered IN. **a** Representative dot plots confirming neutrophil depletion in the lung, with neutrophil gated as single, live, CD45+, Gr1$^{Hi}$Ly6C$^{INT}$. **b** Quantification of lung neutrophils at day 7 p.i. **c** Bacterial burden (CFU/g) of the lung was enumerated at day 7 p.i. Data represent three independent experiments and data are shown as mean ± SD (*n* = 15). Statistical analysis was performed using two-way ANOVA (*$p < 0.01$, ****$p < 0.0001$)

**CD200R$^{-/-}$ neutrophils show aberrant ROS production**. To understand better how CD200R expression regulates neutrophils, we performed RNA-sequencing (RNA-seq) analysis of isolated primary bone marrow WT and CD200R$^{-/-}$ neutrophils. Of the 323 genes that were significantly differentially expressed, one of the major pathways that differed included genes related to ROS clearance and scavenging. The genes upregulated in CD200R$^{-/-}$ neutrophils included haem oxygenase 1 (*Hmox1*), superoxide dismutase 1 (*Sod1*), peroxiredoxin 1 (*Prdx1*), ferritin light polypeptide 1 (*Ftl1*), CDGSH iron sulfur domain 1 (*Cisd1*), peroxiredoxin 4 (*Prdx4*) and thioredoxin reductase 1 (*Txnrd1*) (Fig. 4a). Given the known ability of ROS to target intracellular bacteria in neutrophils[25], we hypothesised that the enhanced bacterial burden observed in CD200R$^{-/-}$ neutrophils may be linked to a reduced ability to produce ROS.

To test whether CD200R$^{-/-}$ neutrophils are defective in their ability to produce ROS, bone marrow neutrophils were isolated from WT and CD200R$^{-/-}$ mice and ROS levels measured via measurement of the uptake and oxidation of the ROS indicator Dihydrorhodamine 123 (DHR123) by flow cytometry. We found that CD200R$^{-/-}$ neutrophils produce significantly less ROS in response to stimulation with PMA than WT neutrophils (Fig. 4b, c). In addition, CD200R$^{-/-}$ neutrophils produced significantly less ROS compared with WT neutrophils in response to *F. tularensis* infection (Fig. 4d, e). Interestingly, production of ROS in WT macrophages was equivalent to that in CD200R$^{-/-}$ macrophages, suggesting that ROS production in these cells was not regulated by CD200R signalling (Supplementary Fig. 5A, B). These findings suggest that lack of CD200R expression in

neutrophils reduces their ability to produce ROS, providing a mechanism by which CD200R deficiency may contribute to enhanced bacterial burdens during *F. tularensis* infection.

## Discussion

Infection with *F. tularensis* can cause devastating illnesses in humans even at very low doses, but mechanisms that contribute to regulation of host immune responses are not fully understood. Here, our results show that CD200R plays an important protective role during pulmonary *F. tularensis* infection, with an increased bacterial burden observed in the absence of CD200R both in vitro and in vivo, and reduced burden observed when the pathway is stimulated by CD200 ligand. Enhanced bacterial burden in the absence of CD200R signalling was dependent on neutrophils, with CD200R$^{-/-}$ mice showing increased neutrophil influx, which likely contributed to increased immunopathological damage as well as providing an enhanced niche for infection. Furthermore, we show that neutrophils from CD200R$^{-/-}$ mice display a reduced ability to produce ROS, an effector function critical in defence against microbial pathogens[25]. Together, our results suggest that in addition to its role in suppression of immune responses, CD200R plays an important role in promoting immunity to infection through modulating neutrophil effector function (see Fig. 5 for summary).

Although it is clear that neutrophils play a role during *F. tularensis* infection, whether they are critical for defence against infection remains unclear, and appears to depend on the route of infection. Neutrophils appear critical in response to a primary systemic infection, with neutrophil-depleted mice administered *F. tularensis* via intravenous or intradermal routes succumbing to normally sub-lethal infection[26,27]. In contrast, the effect of neutrophil depletion during intranasal *F. tularensis* infection (a more natural route of infection) is much less evident, with our data and others[27] showing that neutrophil depletion in WT mice had no effect on pulmonary bacterial burden. These data indicate that neutrophils play a redundant role in dealing with infection in WT mice. However, in situations where the functional capacity of the neutrophils is compromised, such as in the absence of CD200R expression, neutrophils can act as a niche for bacteria during infection (Fig. 5). Interestingly, a recent study suggests that depletion of neutrophils at day 3 p.i., as opposed to at the time of infection, renders mice highly susceptible to pulmonary *F. tularensis* infection[28]. These data suggest that depleting neutrophils at a time when they normally act as a major infected cell type leads to an exacerbated infectious outcome. Thus, together, ours and other studies highlight the fine balance involved in neutrophil-mediated protection.

Unlike other intracellular pathogens, *F. tularensis* is able to escape the neutrophil phagosome, disrupt respiratory bursts and replicate in the cytosol[29,30]. Nevertheless, ROS is still an important mechanisms involved in killing *F. tularensis*. Susceptibility to *F. tularensis* infection is significantly increased in p47$^{phox-/-}$ mice, a model unable to produce respiratory bursts[31], with increased bacterial burdens in the lung and decreased survival[32]. Therefore, if neutrophils provide a safe, replicative niche for *F. tularensis*, we hypothesise that additional dampening of neutrophil responses through the lack of CD200R signalling further exacerbates infectious outcome.

CD200R has been implicated in multiple signalling pathways such as p38 mitogen-activated protein kinase, nuclear factor-κB and ERK, which can have numerous downstream effects[33]. Thus, the changes within neutrophils that lack CD200R could influence an array of signalling pathways, ultimately leading to heightened regulation of ROS production and a protective niche for *F. tularensis* replication. In addition, it is possible that in the absence

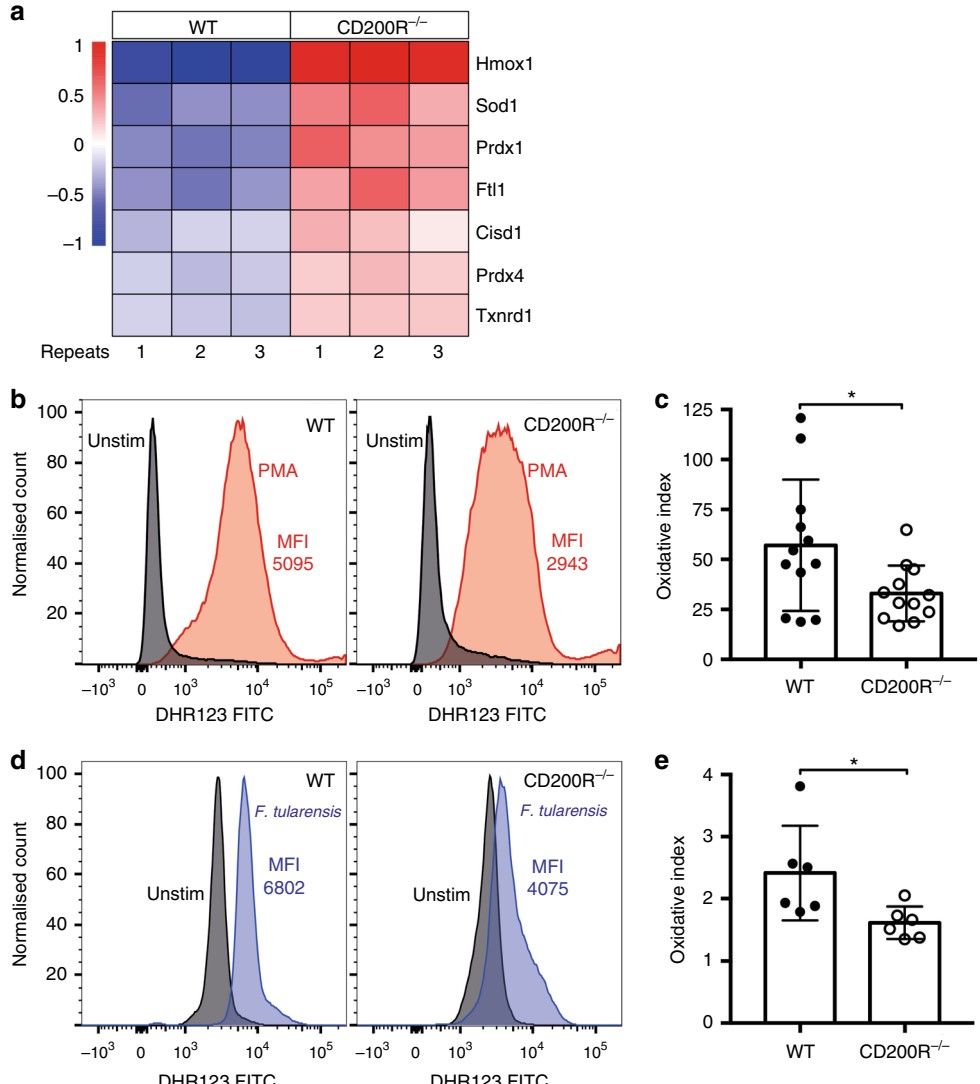

**Fig. 4** CD200R$^{-/-}$ neutrophils show enhanced expression of proteins associated with reactive oxygen species (ROS) clearance and reduced ROS output. Neutrophils were isolated by negative selection from WT and CD200R$^{-/-}$ C57BL/6 mouse bone marrow. **a** Heatmap of selected genes identified in RNA-seq analysis of primary neutrophils isolated from 3 WT and 3 CD200R$^{-/-}$ mice. Signal intensity represents gene expression change following mean centring scaling. **b**, **c** WT and CD200R$^{-/-}$ neutrophils were incubated for 15 min with Dihydrorhodamine 123 (DHR123) (25 ng/ml), followed by 1 h stimulation with Phorbol 12-myristate 13-acetate (PMA) (100 ng/ml). **b** Representative histograms of FITC DHR123 expression in unstimulated or PMA-treated WT and CD200R$^{-/-}$ neutrophils. **c** Oxidative index was quantified using the following equation: mean fluorescence intensity (MFI) Stimulated/MFI Unstimulated. Alternatively, neutrophils were infected with *F. tularensis* MOI 100 and incubated with DHR123 for 24 h. **d** Representative histograms of FITC DHR123 expression in unstimulated and *F. tularensis*-infected neutrophils. **e** Oxidative index was quantified using the following equation: MFI *F. tularensis*-infected/MFI unstimulated. Data represents **b**, **c** four or **d**, **e** two independent experiments and is shown as mean ± SD ($n = 12$ and 6, respectively). Statistical analysis was performed using unpaired *t*-test (*$p < 0.01$)

of CD200R signalling, other non-ROS-mediated pathways may also contribute to the enhanced bacterial burden observed in CD200R$^{-/-}$ mice. Future work will aim to further elucidate the signalling pathways involved in ROS regulation downstream of CD200R and any ROS-independent pathways involved in regulation of responses to *F. tularensis* infection by the CD200R pathway.

In addition, our findings support the idea that removing CD200R signalling in neutrophils could alter myeloid cell dynamics[34,35]. We see enhanced numbers of inflammatory neutrophils during *F. tularensis* infection in the absence of CD200R expression. Despite this enhanced neutrophil presence, it appears that this increased response is not sufficient and in fact leads to increased bacterial burdens.

Other pathogens such as *Leishmania amazonensis* and *Salmonella enterica* have been shown to modulate the CD200/CD200R pathway to increase virulence[36,37]. However, this modulation is usually through increased expression of CD200/CD200R to dampen immunity. Thus, it is of interest that lack of CD200R expression is beneficial to *F. tularensis* virulence and further work is required to determine the role of the CD200/CD200R pathway in responses to different bacterial and other pathogenic infections. In addition, it will be important in the future to address the role of CD200R signalling during infection with type A strains of *F. tularensis*, which show greater virulence in humans than *F. tularensis* LVS.

Our data therefore suggest an important role for CD200R signalling in controlling pathogens, particularly within

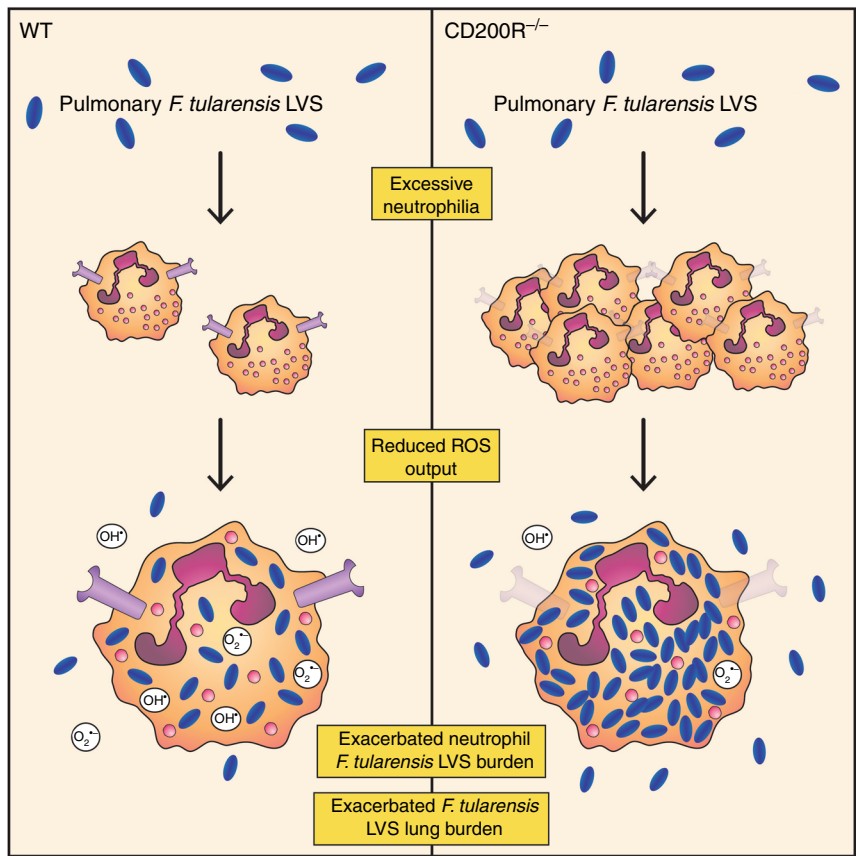

**Fig. 5** Lack of CD200R in mice causes exacerbated pulmonary *F. tularensis* infection via neutrophil-dependent mechanisms. Pulmonary challenge with *F. tularensis* LVS causes an excessive neutrophil influx in the lung of CD200R$^{-/-}$ mice compared with WT. Due to a reduced ROS output from CD200R$^{-/-}$ neutrophil, they provide a replicative niche for *F. tularensis* LVS, leading to an increased burden in CD200R$^{-/-}$ neutrophils. This ultimately leads to an increased overall *F. tularensis* LVS burden in the lung of CD200R$^{-/-}$ mice compared with WT

neutrophils. Our findings of dysfunctional ROS production in CD200R$^{-/-}$ neutrophils highlights the need for further investigation into the role of CD200R in the antimicrobial activity toward intracellular pathogens such as *F. tularensis*. Neutrophils are clearly a double-edged sword, with the cells' contribution to the resolution of infection in the context of a complex host environment found during acute, highly infectious diseases, a fine line between promoting destruction of the bacteria and acting as an intracellular niche[38]. Nevertheless, based upon the evidence presented here, manipulating neutrophils through the modulation of the CD200R pathway could potentially be a novel therapeutic target for the treatment of infections caused by *F. tularensis*.

## Methods

**Bacterial strains**. *F. tularensis* LVS was used in all experiments and was derived from an original NDBR101 *Pasteurella tularensis* live vaccine, experimental lot 4 produced during the 1960s, stored at −20 °C in the culture collection at Defence Science and Technology Laboratory (Dstl, Porton Down). Bacterial inoculum was prepared by growing on blood cysteine glucose agar (BCGA) at 37 °C for 48 h. *F. tularensis* was then resuspended in phosphate-buffered saline (PBS) to an optical density at 600 nm of 0.20, which corresponds to ~1 × 10$^9$ CFUs/ml. The required CFUs were then obtained through further dilutions.

**Generation of BMDMs**. Hind legs were removed from WT and CD200R$^{-/-}$ mice, and bone marrow flushed from bones using sterile PBS. After red blood cell lysis, cells were resuspended in complete media (RPMI: 20% fetal calf serum (FCS), 20 mM HEPES, 20 ng/ml macrophage colony-stimulating factor (M-CSF)) and 6 × 10$^6$ cells were seeded in a 10 cm non-treated culture dish. Cells were collected and used after 7 days in culture.

**Primary neutrophil isolation**. Bone marrow was isolated from hind legs of WT and CD200R$^{-/-}$ mice, and cells incubated with a neutrophil isolation biotin-antibody cocktail (Miltenyi Biotec) for 10 min at 4 °C, washed with wash buffer, then incubated for 15 min at 4 °C with anti-biotin microbeads. Neutrophils were isolated by magnetic separation with MS MACS columns and a MACS Separator (Miltenyi Biotec).

**In vitro *F. tularensis* infection assay**. *F. tularensis* inoculum was prepared as described previously and multiplicity of infection (MOI) of 100 achieved through serial dilutions in L15 media (10% FCS, 5 mM L-Glutamine) (Life Technologies). Actual inoculum MOI was determined by plate count on BCGA. Cells were incubated for 2 h (BMDMs) or 4 h (primary neutrophils) with *F. tularensis* inoculum at 37 °C, to allow for cellular uptake of the bacteria. Medium was removed and cells were washed with warm PBS, followed by 30 min incubation with 10 µg/ml gentamicin (Sigma) to kill any extracellular bacteria. To determine bacterial load at desired time points, cells were lysed for 2–3 min with 4 °C H$_2$O, while scraping, cell dilutions of lysates plated onto BCGA for 4–5 days and single colonies counted to determine *F. tularensis* CFU/ml.

**Mice and in vivo *F. tularensis* infection**. Female C57BL/6 mice (Charles River, UK) and CD200R$^{-/-}$ mice, developed on a C57BL/6 background[39], were kept in specific pathogen-free conditions according to institutional and UK Home Office guidelines in the Biological Services Unit at The University of Manchester. Eight- to 10-week-old mice were infected intranasally with 50 µl PBS containing ~1000 CFU *F. tularensis*. Actual infection dose in each experiment was determined by plate count on BCGA. All procedures were performed in accordance with the Home Office Scientific Procedures Act (1986) and under the DERFA license. The humane endpoint for all experiments was defined as weight loss in excess of 25% of starting body weight and/or loss of mobility.

**Enumeration of bacterial burden**. Numbers of CFU in the whole left lobe of the lung-infected mice were determined by pressing organs in PBS containing protease inhibitor cocktail (Roche) through a 40 µM filter. Organ homogenates were plated

on BCGA, incubated at 37 °C for 4–5 days, single colonies counted and final CFU/g calculated using corresponding organ weight.

**Digestion of lung tissue**. Lung tissue was chopped into small pieces, placed in Hank's buffered salt solution (HBSS) containing 100 µg/ml Liberase TM (Roche) and 100 µg/ml DNase I (Roche), then left to shake at 37 °C for 30 min. HBSS containing 5 mM EDTA was added to stop the digest. Digested tissue was then passed through a 40 µM sieve before lysis of red blood cells and washing in PBS.

**Flow cytometry**. Cells were first stained with LIVE/DEAD® Fixable Blue Dead Cell Stain kit (Invitrogen), washed in fluorescence-activated cell sorting buffer (1% bovine serum albumin, 0.5% sodium azide in 1× PBS) then incubated with FcR blocking CD16/32 antibody (eBiosciences) at 5 µg/ml for 20 min at 4 °C. Subsequently, cells were stained with antibodies at 1 µg/ml directed against CD45 (clone 30-F11, #103138), Gr1 (clone RB6-8C5, #108406), I-A/I-E (clone M5/114.15.2, #107622), CD11b (clone M1/70, #101226), Ly6C (clone HK1.4, #128032), TCR-β (clone H57-597, #109241), CD4 (clone RM4-5, #100546), Ly6G (clone 1A8, #108443), CD8 (clone 53-6.7, #100750), CD200R (clone OX-110, #123908), CD64 (clone X54-5/7.1, #139314) (Biolegend), Siglec-F (clone 1RNM44N, #46-1702-82) and CD11c (clone N418, #61-0114-82) (eBiosciences) for 30 min at 4 °C. For intracellular staining, cells were resuspended in Fixation/Permeabilisation buffer (concentrate:diluent, 1:3) (eBioscience) and incubated at 4 °C overnight. Cells were then washed in Permeabilisation buffer (eBioscience), centrifuged and incubated with anti-*F. tularensis* lipopolysaccharide (LPS) antibody (clone FB11, Thermo #MA1-21690) labelled with APC Conjugation Kit (Abcam) in permeabilisation buffer (eBioscience) for an additional 30 min at 4 °C. Cells were analysed on a LSR Fortessa (BD Biosciences). Data were further analysed using FlowJo software (Tree Star, Ashland, Oregon).

**Antibody-mediated neutrophil depletion**. WT and CD200R$^{-/-}$ mice were treated intraperitoneally with 50 µg *InVivo*Plus anti-mouse Ly6G (clone 1A8) (Bio X Cell, #BP0075-1) or *InVivo*Plus rat IgG2a isotype control (Bio X Cell, #BP0089) at day −1, 1, 3 and 5 of *F. tularensis* intranasal infection.

**CD200-Fc treatment**. Female C57BL/6 mice were treated intranasally with 10 µg CD200-Fc (#3355-CD-050, R&D Systems) or human IgG (#110-HG-100, R&D Systems) in 50 µl PBS at day 3 post-intranasal challenge with 1000 CFU *F. tularensis* LVS.

**Measurement of ROS production using DHR123**. WT and CD200R$^{-/-}$ primary neutrophils were incubated for 15 min with DHR123 (25 ng/ml), followed by 1 h stimulation with PMA (100 ng/ml). Alternatively, primary neutrophils were infected with *F. tularensis* MOI 100 and incubated with DHR123 for 24 h. Cells were labelled with anti-Ly6G to identify neutrophils and fluorescein isothiocyanate DHR123 expression then measured by flow cytometry using a BD FACSCanto II. Oxidative index was quantified using the following equation: mean fluorescence intensity (MFI) Stimulated/MFI Unstimulated. DHR ratio was quantified using the following equation: MFI DHR+/MFI DHR−.

**RNA-seq analysis and heatmap generation**. RNA was extracted from bone marrow neutrophils WT and CD200R$^{-/-}$ mice. Total RNA was submitted to the Genomic Technologies Core Facility (University of Manchester) and quality/integrity of samples assessed using a 2200 TapeStation (Agilent Technologies). Libraries were generated using the TruSeq® Stranded mRNA assay (Illumina, Inc.) according to the manufacturer's protocol. Briefly, total RNA (0.1–4 µg) was used as input material from which polyadenylated mRNA was purified using poly-T, oligo-attached, magnetic beads. The mRNA was then fragmented using divalent cations under elevated temperature and then reverse transcribed into first-strand complementary DNA using random primers. Second-strand cDNA was then synthesised using DNA Polymerase I and RNase H. Following a single 'A' base addition, adaptors were ligated to the cDNA fragments and the products then purified and enriched by PCR to create the final cDNA library. Adaptor indices were used to multiplex libraries, which were pooled prior to cluster generation using a cBot instrument. The loaded flow cell was then paired-end sequenced (76 + 76 cycles, plus indices) on an Illumina HiSeq4000 instrument. Finally, the output data were demultiplexed (allowing one mismatch) and BCL-to-Fastq conversion performed using Illumina's bcl2fastq software, version 2.17.1.14.

Read trimming was carried out with Trimmomatic[40] and then quality control was validated with FastQC software (Babraham Bioinformatics). Pseudoalignment was performed with Kallisto to generate transcript abundance estimates[41]. Tximport[42] was applied to create a gene-level count matrix that was further analysed with DESeq2 package[43] to perform differential expression analysis. Finally, an unsupervised gene clustering analysis was applied with the pheatmap package (https://cran.r-project.org/web/packages/pheatmap/index.html) using a mean centring scaling.

**Statistical analysis**. All graphs and statistical analysis were produced using GraphPad Prism 6. Data were expressed as the mean ± SD. Normality was determined using the D'Agostino–Pearson and Shapiro–Wilk normality tests. Homogeneity of variance was assessed by Brown–Forsyth and F tests, with Brown–Forsyth and Welch's corrections made where appropriate. Statistical analysis was performed by two-way analysis of variance and Tukey's multiple comparison test, unpaired *t*-tests or Mantel–Cox test. Significance was considered at $p < 0.05$.

**Reporting summary**. Further information on research design is available in the Nature Research Reporting Summary linked to this article.

## Data availability
The datasets generated during and/or analysed during the current study are available from the corresponding author on reasonable request. Raw data associated with Figs. 1b, d, 2a–j, 3b, c, 4a, c, e and Supplementary Figs. S1A-D, S2A-B, S2D-E, S4A-F and S5B are available via a source data file submitted with this manuscript. RNA-seq generated as part of Fig. 4 has been deposited in GEO Bank (accession number GSE129287).

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

## Acknowledgements

We thank the members of the Travis lab and the Defense Science and Technology Laboratory for helpful discussions and advice. We thank Dr. Gareth Howell and the University of Manchester Flow Cytometry Core Facility for facilitating flow cytometry analysis, the Genomics Technologies and Bioinformatics Core Facilities for aid in RNA-sequencing, and members of the Biological Service Facility at the University of Manchester for help with animal work. This work was supported by funding from the United Kingdom Ministry of Defense, the Defense Science and Technology Laboratory. M.A.T. was supported by funds from the Defense Science and Technology Laboratory, and the Medical Research Council (grant number MR/M00242X/1), T.H. was supported by funds from the Wellcome Trust (grant number 02865/Z/16/Z). The Wellcome Centre for Cell-Matrix Research, University of Manchester, is supported by core funding from the Wellcome Trust (grant number 203128/Z/16/Z).

## Author contributions

J.C., M.E.F., S.A.H. and J.D. performed experiments. S.R. performed RNA-seq analysis. J.C., M.A.T., R.V.D. and T.H. designed and supervised the study. J.C. and M.A.T. wrote the manuscript. R.V.D., E.D.W. and G.C. provided study advice and manuscript editing.

## Additional information

**Competing interests:** The authors declare no competing interests.

