## [Peer Review File · Nature Communications]

Reviewers' comments:

Reviewer #1 (Remarks to the Author):

Casulli et al compared the impact of airway exposure to *Francisella tularensis* between WT and CD200R1-deficient mice. They claim that CD200R1 deficiency leads to an increased infectious burden in the lung, which is caused by a defective ROS generation by neutrophils. As a result, CD200R deficiency would allow an increased FT accumulation in neutrophils, which would account for the increased FT burden in the lung.

A number of ex vivo and in vivo experiments were performed. First, the accumulation of FT was compared ex vivo between Neutrophils and BMDM from WT or CD200R1-deficient mice. In vivo, the frequency of FT-positive leukocyte subsets was assessed, as well as the overall pulmonary infectious burden, while no indicators of diseased severity were presented. The impact of neutrophil depletion was also compared between WT and KO mice regarding the overall bacterial burden, although a neutrophil-specific knock out would have probably served better the conclusions of the study. Finally, the authors present evidence that the expression of numerous genes relating to the generation of ROS are downregulated in neutrophils isolated from CD200KO vs and WT mouse. It remains unclear how many times this experiment was repeated. Thereafter, the authors attempt to gather functional data on the generation of ROS in WT and KO cells in order to strengthen the concept that this deregulation explains the increased FT burden.

The overall findings are intriguing given the need to enhance our understanding of the mechanisms supporting extreme virulence of FT. Moreover, the CD200-CD200R regulatory pathway is complex, misunderstood, and significant strides are required to better appreciate its regulation, as well as its pre-clinical potential. For that matter, the finding that a knock out for an immunoregulatory molecule promotes infection is highly surprising and interesting. The authors conclude that manipulating CD200R, presumably with agonists, could provide useful in the context of FT infection. Yet critical experiments are missing to strongly support this contention.

Although the findings are interesting, there is a number of clarifications that need to be brought to this manuscript before the claims and conclusions can be fully supported. Moreover, the mechanistic link between CD200R and the regulation of ROS is a piece that is critically missing in this manuscript in order to make it field-changing.

Below is a list of items that require attention in order to clarify to which extent the proposed manuscript could yield a significant advance in the field of immunology.

Major comments:

There is a number of questions that should be addressed before clear conclusions can be drawn from Figure 1. First, it is well accepted that CD200R1 activity is linked with the availability of its ligand. Yet, the first series of experiments is performed without addition of the soluble ligand ex vivo, and the presence of CD200 at the surface of the cultured cells was not assessed. Are the authors claiming for an intrinsic activity of CD200R1? In view that cells appear to be altered phenotypically, the authors should make a better case that the mice were either bred het x het or that the background was recently refreshed.

In addition, the frequency of infected cells could be influenced by differing cell mortality between experimental conditions (especially for the 24h time point). Thus the viability of the cells should be addressed.

Figure 2: There also is a concern regarding the units used to compute the bacterial burden (CFU per gram of tissue). This is because the magnitude of the lung pathology is often associated with a modified water content, which can become a confounding factor when comparing groups. It is usual to monitor the burden in a whole lobe or organ to become independent of the altered water content of the tissue. Thus providing the critical details of the method is a prerequisite for the readership to better comprehend the data.

The conclusion that the increased burden in vivo is limited to neutrophils also requires better investigation. This is because the authors do not account for the exit of CD11b-positive cell subsets towards the draining lymph nodes, although it has been documented. How does their gating strategy exclude dendritic cells?

Figure 4: The question as why the cells isolated from KO mice and cultured ex vivo display an altered phenotype remains open. In face of strong evidence that germline deletion can lead to adaptative mechanisms in growing animals, the presence/modulation of the ligand and/or other CD200 receptors should have been addressed. This also applies for the in vivo experiments. In fact, one important missing piece of information in this manuscript is how does the absence of CD200R leads to altered ROS generation? There are subsets of CD200 receptors whose interplay and functions are increasingly recognized in mice. The unacknowledged complexity of this system is a weakness regarding the understanding of the CD200-CD200R receptor pathway, which decreases the impact of this manuscript.

Figure 4 : It is unclear from how many individual mice Panel A analyses were made. In panel C, because the level of autofluorescence is often modulated in the 450-550 nm spectrum between cells that are differentially activated, the level of autofluorescence should be subtracted (without DHR123; for each experimental condition) before computing the ratio. Panels D-F also raise more questions. For instance, we ignore the proportion of FT-positive neutrophils in the DHR+ vs DHR-cells. This is a major issue because the proportion of infected cells is changed in WT vs KO mice in figure 1. Having the DHR123 data for macrophages would also have potentially contributed to better explain the differences in terms of FT burden between both cell types in vivo. Finally the authors claim that altered ROS generation accounts for an increased burden of FT in neutrophils. Yet, definite proof that FT viability is enhanced in neutrophils from CD200R-deficient mice is never provided.

Regarding supplementary figure 1, a reference should be provided to support that the claimed cell subsets are accurately identified using the proposed gating strategy. Otherwise, cytopins followed by a GEMSA-like staining should be performed to identify the classical populations seen in the lung (alveolar macrophages, vs neutrophils, vs eosinophils & lymphocytes).

In the discussion the authors write that their results support that CD200R impacts on myeloid cell subset migration, after which they state that absolute numbers of inflammatory cells are increased to deal with FT infection in the lung. If the data regarding absolute numbers of cells is available, it would certainly benefit the validity of the manuscript, especially in the context where the authors intend to make a point that supernumerary neutrophils act as an additional reservoir for FT in the KO mice; which is difficult to support using frequencies, as it is currently the case. In addition, the notion that CD200R impacts on migration should be better substantiated. How can it be differentiated from recruitment in the context of your study?

CD200R ligands exist for in vitro and in vivo use. This type of approach would have served the concluding sentence of the manuscript much better. In the current form this concluding remark remains highly speculative.

Statistics should also state how the assumption of normality and equal variance between groups were assessed prior to choosing parametric tests. This is critical to provide the readership with the insurance that the statistical differences seen at figure 2 A are accurate.

Additional comments:

Although ROS contribute to the neutralization of FT, this pathogen is considered to be somehow resistant to the respiratory burst. In spite of the PMA results, it remains difficult to reconcile whether the altered Dihydrorhodamine123 signal results from an altered function of CD200R1-deficient cells, or from their increased FT burden. This should be addressed before concluding in an intrinsic defect in ROS productions in CD200R1-deficient cells upon infection. For instance, is there a loss of difference between the groups in the presence of ROS inhibitors?

Reviewer #2 (Remarks to the Author):

In this study, the authors demonstrate that CD200 receptor (CD200R) is involved in the control of *F. tularensis* infection by neutrophils, both in vitro and in vivo in a mouse pneumonia model. Deletion of CD200R results in higher proliferation of bacteria inside neutrophils, which therefore become a niche for *F. tularensis* replication. This work provides the first evidence that the role of CD200R may not be limited to dampening immune responses in the infected host, but may also participate in the control of infection by phagocytes. This would open potential new therapeutic perspectives for infections caused by intracellular pathogens.

Major comments.

F. tularensis is one of the few pathogens able to resist the harsh conditions found in human neutrophils. Moreover, it has been demonstrated that this bacterium may increase the lifespan of these phagocytic cells so as to use them as a replication niche during the initial phase of host infection. The demonstration that CD200R may be involved in the control of *F. tularensis* infection by neutrophils is very interesting. However, the authors should better describe the impact of CD200R deletion on the rate of progression and the severity of infection in their murine model. Does the increased multiplication of *F. tularensis* in neutrophils really affect host resistance to infection? They should also further describe the potential clinical relevance of their findings in patients suffering from severe pneumonic tularemia. Are there known deficiencies of the CD200 / CD200R pathway in humans? How can one really consider modulating the CD200/CD200R response to improve the prevention of these serious and often rapid infections?

Minor comments.

- Line 60. "however infection via the respiratory route is most virulent,". The bacterium may not be more virulent when infection occurs via the respiratory route. Pneumonic tularemia caused by type A strains of *F. tularensis* is usually characterized by a rapid onset and evolution to a life-threatening infection. However, several reasons may explain higher severity of such infections: higher bacterial inoculum (compared to skin inoculation via tick bites for example); less effective immune control; etc.
- Line 108. The *F. tularensis* LVS strain is a type B type (subsp. *holarctica*) strain of *F. tularensis*. Its virulence is highly attenuated in humans, although it remains highly virulent in mice. Moreover, in humans, severe pulmonary infections are usually caused by type A (subsp. *tularensis*) strains of *F. tularensis*, while type B strains usually cause subacute or chronic pneumonia. Therefore, the LVS strain was not the most appropriate model for the present study. The authors should discuss this limitation in the appropriate section.
- Line 136. "cells were lysed for 2-3 mins with cold H₂O". This is not the most common cell lysis methodology used before CFU enumeration. Did authors check that all phagocytic cells were lysed by this procedure? Please specify which temperature was used. A reference should be added.
- Line 140. "Female C57BL/6 mice (Charles River, UK) and CD200R^{-/-} mice, developed on a C57BL/6 background". Are these two models strictly comparable except for CD200R deletion?
- Line 143. "8-10 week old mice were infected intranasally (IN) with 50 μ l PBS containing ~1000 CFU *F. tularensis*." Why such a high bacterial inoculum as used for these experiments, while only 10-100 CFU are usually needed to kill all infected mice?
- Line 273. "Bacterial burden in the lung showed no significant differences at early time points p.i. (day 1, 3 or 5 p.i.) (Fig. 2A). However, we saw a significantly enhanced bacterial burden at day 7 p.i. in CD200R^{-/-} mice (Fig. 2A)." The authors should further discuss the significance of this finding. How can they explain such a late effect of CD200R defect on *F. tularensis* infection in mice? In the observed difference clinically relevant considering that acute pneumonia in humans may be lethal within 3-5 days after disease onset?
- Line 280. "Interestingly, we found that the increased bacterial burden in CD200R^{-/-} mice was restricted to neutrophils, alongside a dramatic influx of neutrophils into the infected CD200R^{-/-} lung at days 5 and 7". How can authors explain that bacterial burden was not altered in macrophages, although these cells also express CD200R? More importantly, *F. tularensis* has been

reported to enhance lifespan of infected neutrophils. How could authors differentiate increased neutrophil lifespan from neutrophil influx in infected CD200R^{-/-} mice?

- Line 299 "Depletion of neutrophils in CD200R^{-/-} mice rescued the bacterial burdens so that it was comparable to those in WT mice (Fig. 3C)." Previous experiments have shown that depletion of neutrophils in *F. tularensis*-infected mice could reduce pathological effects associated with lung infection. Was there any difference between WT and CD200R^{-/-} mice when both were depleted in neutrophils?

- Lines 342-345. The authors should further discuss if higher bacterial burden in neutrophils from CD200R^{-/-} mice were associated with worse pathological findings in infected lung tissues, or higher death rates in the corresponding mice compared to controls? In other words, was *F. tularensis* infection much more severe in CD200R^{-/-} mice?

- Line 393. "Nevertheless, based upon the evidence presented here, manipulating neutrophils through the modulation of the CD200R pathway could potentially be a novel therapeutic target for the treatment of infections caused by *F. tularensis*." There are examples of other intracellular human pathogens that modulate the CD200/CD200R pathway to increase their virulence (e.g. *Mycobacterium tuberculosis*, *Brucella* sp.). These examples should be discussed. Could *F. tularensis* modulate this cell pathway as well? Also, a number of molecules have been already used to modulate this CD200/CD200R pathway. Could these compounds be tested in the *F. tularensis* model?

Response to reviewers, Casulli et al.:

We thank both reviewers for their time in reviewing our manuscript, and for providing supportive and constructive comments. Below we present our responses to the questions raised and highlight where new data and text has been added to the manuscript to address the points raised.

Reviewer 1

There is a number of questions that should be addressed before clear conclusions can be drawn from Figure 1. First, it is well accepted that CD200R1 activity is linked with the availability of its ligand. Yet, the first series of experiments is performed without addition of the soluble ligand ex vivo, and the presence of CD200 at the surface of the cultured cells was not assessed. Are the authors claiming for an intrinsic activity of CD200R1?

The reviewer is correct that we do not add exogenous CD200 into our in vitro culture assays in Figure 1. We have now assessed CD200 expression on bone marrow macrophages and neutrophils (cells used in our in vitro assays in Figure 1) by flow cytometry and show that there is low, but detectable expression of CD200 ligand on the surface of both cell types (Figure for Reviewers 1). Interestingly, there appears to be enhanced levels expressed on the surface of CD200R^{-/-} macrophages, but not neutrophils (Figure for Reviewers 1). There is therefore potential for CD200 expression by both macrophages and neutrophils to stimulate CD200R in an autocrine fashion, with this stimulation not possible in CD200R-deficient cells. However, we cannot rule out contribution of other CD200 ligand cellular sources in vivo playing a role in signalling to cells prior to their isolation and use in assays.

Figure 1 for reviewers: Analysis of CD200 expression on WT and CD200R^{-/-} bone marrow-derived macrophages and neutrophils. Primary BMDM were derived from WT and CD200R^{-/-} mice and A) CD200 expression and B) CD200 MFI was measured by flow cytometry. C) Representative histogram of CD200 fluorescence intensity in WT (red) or CD200R^{-/-} (blue) BMDM, with control (black). Neutrophils (CD45⁺CD11b⁺Ly6G⁺) were gated within bone marrow from WT and CD200R^{-/-} mice and D) CD200 expression and E) CD200 MFI was measured by flow cytometry. F) Representative histogram of CD200 fluorescence intensity in WT (red) or CD200R^{-/-} (blue) bone marrow neutrophils, with control (black). Data is n=3, and is shown as mean ± SD. Statistical analysis was performed using unpaired t-tests (***) p<0.001).

In view that cells appear to be altered phenotypically, the authors should make a better case that the mice were either bred het x het or that the background was recently refreshed.

The reviewer is correct that there is the potential for any mouse strain to develop some genetic drift. Therefore, from data presented in our original submission we cannot definitively rule out that differences between CD200R^{-/-} mice and C57BL/6 controls were not due to some genetic drift between control and KO animals. However, we have now performed additional experiments manipulating the CD200R pathway in wild type C57BL/6 mice to directly determine if this pathway does play an important role in controlling infection with Francisella infection.

*Thus, we now demonstrate that stimulating the CD200R pathway by administration of a single intranasal treatment of CD200-Fc (a ligand for CD200R, Gorczynski et al 1999, PMID: 10415071) at day 3 p.i. (coinciding with onset of weight loss) significantly reduced the bacterial burden in the lung at day 7 p.i. with *F. tularensis* compared to human IgG control-treated WT mice, with a greater than 50% reduction achieved (Figure 2H and I in new manuscript and new text lines 249-254). Additionally, decreased neutrophil influx was observed in CD200-Fc-treated mice (Figure 2J in new manuscript and new text lines 254-255). These data therefore fully support the results obtained in the absence of CD200R in mice: in the absence of the pathway we see exacerbated infection and enhanced neutrophil influx, whereas when we stimulate the pathway in WT mice we observe less infection and neutrophil influx. Thus, we feel it is very unlikely that the phenotypes observed in the absence of CD200R are due to genetic drift of this strain.*

In addition, the frequency of infected cells could be influenced by differing cell mortality between experimental conditions (especially for the 24h time point). Thus the viability of the cells should be addressed.

*We now provide additional data analysing the cell viability of isolated neutrophils and BMDM at the 24h time point (Figures S1C and 1D respectively). These data show that there are no significant differences in cell viability of WT and CD200R^{-/-} BMDM and neutrophils, either uninfected or *F. tularensis*-infected. Thus, the increased bacterial burden displayed in Figure 1B and D is very likely independent of cell viability. We have added additional text to the manuscript to describe the new supplementary figures (lines 209-212).*

Figure 2: There also is a concern regarding the units used to compute the bacterial burden (CFU per gram of tissue). This is because the magnitude of the lung pathology is often associated with modified water content, which can become a confounding factor when comparing groups. It is usual to monitor the burden in a whole lobe or organ to become independent of the altered water content of the tissue. Thus providing the critical details of the method is a prerequisite for the readership to better comprehend the data.

We apologise for the lack of clarification in describing the enumeration of bacterial burdens in the lung. The whole left lobe of mouse lungs was used when monitoring bacterial burden during infection. The whole left lobe was weighed and burden represented as CFU/g to normalise for any differences in the amount/weight of tissue taken. Additional text has been added in the Materials & Methods to clarify this point (line 127).

However, to directly address the reviewers point about potential differences in bacterial burdens being due to altered water content (and therefore weight) in control versus $CD200R^{-/-}$ mice, we assessed bacterial burden in whole lung lobes from control and $CD200R^{-/-}$ mice without correcting for weight of the organ. We find that there is still a significant increase in bacterial burden in the lung of $CD200R^{-/-}$ mice without correcting for organ weight (Figure 2A for reviewers). Indeed, we do not find any significant difference in lung wet weight after infection in control versus $CD200R^{-/-}$ mice (Figure 2B for reviewers). Thus, the differences in bacterial burdens observed between control and $CD200R^{-/-}$ mice are not due to normalising burden to lung weight.

Figure 2 for reviewers: $CD200R^{-/-}$ mice show significantly increased *F. tularensis* burden in the lung when CFU is not normalised to lung weight. WT and $CD200R^{-/-}$ C57BL/6 mice were infected IN with a challenge dose of 1000 CFU *F. tularensis* LVS. A) Bacterial burden in WT and $CD200R^{-/-}$ lung at day 7 p.i. presented as raw CFU, not normalised to lung weight. B) Lung lobe weight in WT and $CD200R^{-/-}$ mice at day 7 p.i. Data represents four independent experiments (n=15-18) and is shown as mean \pm SD. Statistical analysis was performed using Mann-Whitney test (*p<0.05).

The conclusion that the increased burden in vivo is limited to neutrophils also requires better investigation. This is because the authors do not account for the exit of CD11b-positive cell subsets towards the draining lymph nodes, although it has been documented. How does their gating strategy exclude dendritic cells?

Our gating strategy to gate neutrophils uses Ly6G which is a neutrophil-specific marker (e.g. Daley et al. 2008, PMID: 17884993), as well CD11b. Thus, we do not expect any other cell types to be present in the CD11b⁺ Ly6G⁺ population we have described as neutrophils (Figure S2). Furthermore, when gating CD11b⁺ Ly6G⁻ macrophage populations we use

CD64 as a marker that has previously been used to distinguish between CD11b⁺ macrophage and dendritic cell populations in the lung (e.g. Misharin et al. 2013, PMID: 23672262).

Following the successful identification of neutrophil and macrophage populations in the lung, CD11b⁺ Ly6G⁻ CD64⁻ populations could be further identified as dendritic cells through the expression of MHC-II and CD11c. During the course of *F. tularensis* infection we saw a significant reduction in the percentage of CD11b⁺ DCs in the lungs of CD200R^{-/-} mice compared to WT at day 1 and 3 p.i. (Figure 3A for reviewers), which as the reviewer suggested could be due to the exit of these cells to the draining lymph node. However, there was no significant difference in proportions of these cells at day 5 and 7 p.i. (the time points when we see enhanced neutrophil frequencies) (Figure 3A for reviewers) and also no difference in the percentage of infected CD11b⁺ DCs during the course of infection (Figure 3B for reviewers). As we also saw no differences in CD11b⁺ macrophage infection levels in the lung (Figure 2G), it is unlikely these other CD11b⁺ populations are contributing to increased *F. tularensis* burden in vivo.

Figure 3 for reviewers: Decreased percentage of CD11b⁺ DCs during early infection in CD200R^{-/-} mice, yet no difference in percentage infection compared to WT mice. WT and CD200R^{-/-} C57BL/6 mice were infected IN with a challenge dose of 1000 CFU *F. tularensis* LVS. A) Quantification of CD11b⁺ dendritic cells (CD45⁺Ly6G⁻CD64⁻MHC-II⁺CD11c⁺CD11b⁺) as a percentage of total CD45⁺ cells at day 1, 3, 5 and 7 p.i. B) Percentage of CD11b⁺ dendritic cells infected with *F. tularensis* at day 1, 3, 5 and 7 p.i. Data is representative of two independent experiments (n=3-4) and is shown as mean ± SD. Statistical analysis was performed using two-way ANOVA (*p<0.05. ***p<0.001).

Figure 4: The question as why the cells isolated from KO mice and cultured ex vivo display an altered phenotype remains open. In face of strong evidence that germline deletion can lead to adaptive mechanisms in growing animals, the presence/modulation of the ligand and/or other CD200 receptors should have been addressed. This also applies for the in vivo experiments.

The reviewer is correct that there could conceivably be alterations in other CD200Rs or CD200 ligand by lack of CD200R (common name for CD200R1), and have performed additional experiments to address these points.

*Thus, we analysed expression of CD200 ligand on bone marrow-derived neutrophils (used in in vitro assays) and show low, but indistinguishable levels of CD200 expression on control versus CD200R^{-/-} cells (Figure 1 for reviewers D-F). We have also analysed expression of CD200 in vivo on CD45-negative cells and also neutrophils and macrophages in the lung during infection. We demonstrate that at day 7 p.i. with *F. tularensis* there was no significant change in the expression of CD200 in CD45-negative cells, neutrophils or interstitial macrophages between WT and CD200R^{-/-} mice (Figure 4A and B for reviewers). These data suggest there is no modulation of ligand expression in the absence of CD200R.*

To determine if expression of CD200R family members was altered on CD200R^{-/-} cells, we examined expression on neutrophils from WT versus CD200R^{-/-} mice. As expected there was no CD200R1 expression detected in CD200R^{-/-} neutrophils (Figure 4C for reviewers), as these mice have specific knockout of this member of the CD200R family (Boudakov et al. 2007, PMID: 17667818). CD200R2 expression was undetectable in both WT and CD200R^{-/-} neutrophils (Figure 4D for reviewers), consistent with data in the Immgen database. CD200R^{-/-} neutrophils show significantly reduced expression of both CD200R3 (Figure 4E for reviewers) and CD200R4 (Figure 4F for reviewers) compared to WT neutrophils. As opposed to the inhibitory activity of CD200R1, CD200R3 and 4 have been proposed to be activating receptors due to their important interactions with DAP12 (Wright et al. 2003, PMID: 12960329). Thus, these data suggest there is potential modulation of CD200R3 and 4 expression in the absence of CD200R1. However, the opposing roles of CD200R1 and CD200R3 and 4 brings into question the contribution, if any, of decreased expression of these activating receptors towards the enhanced infectious phenotype observed in CD200R^{-/-} mice.

Figure 4 for reviewers: No modulation of CD200 in the lung of CD200R^{-/-} mice, yet decreased gene expression of the activating CD200R3 and CD200R4 in CD200R^{-/-}-derived neutrophils. A) Percentage of CD200-expressing and B) CD200 MFI in neutrophils, CD11b⁺ macrophages and CD45-cells in the lung of WT and CD200R^{-/-} mice at day 7 p.i. with *F. tularensis*. C-F) Quantitative real-time PCR of CD200R1, CD200R2, CD200R3 and CD200R4 mRNA expression in WT and CD200R^{-/-} neutrophils. Results are presented as expression fold change in relation to expression in WT neutrophils following normalisation to HPRT mRNA levels. Data represents one (A and B) or two (C-F) independent experiments and data is shown as mean \pm SD (n=5-6). Statistical analysis was performed using two-way ANOVA (A-B) and unpaired t-tests (C-F) (*p<0.05, **p<0.01).

In fact, one important missing piece of information in this manuscript is how does the absence of CD200R leads to altered ROS generation? There are subsets of CD200 receptors whose interplay and functions are increasingly recognized in mice. The unacknowledged complexity of this system is a weakness regarding the understanding of the CD200-CD200R receptor pathway, which decreases the impact of this manuscript.

The reviewer is correct that the signalling pathways and mechanisms by which CD200R leads to altered ROS generation is an important question. However, the functions downstream of CD200R that are responsible for ROS generation are likely very complex and will take considerable additional experiments and data to decipher. Thus, we feel that elucidating these mechanisms is out-with of this current study and should be the subject of future work. We have added additional text to the manuscript (lines 349-353) to highlight this point.

Figure 4: It is unclear from how many individual mice Panel A analyses were made.

We apologise for not making this clearer in the figure legend. Panel A is made up of neutrophils isolated from 3 WT and 3 CD200R^{-/-} mice. The figure has been annotated accordingly and text has been added to the figure legend to make this clear.

In panel C, because the level of autofluorescence is often modulated in the 450-550 nm spectrum between cells that are differentially activated, the level of autofluorescence should be subtracted (without DHR123; for each experimental condition) before computing the ratio.

*We thank the reviewer for the suggestion and have made the appropriate changes to the analysis. The conclusions in the original submission (decreased ROS production by CD200R^{-/-} neutrophils) are still fully supported, with a significant reduction observed when cells are stimulated with PMA (Figure 4B-C) or infected with *F. tularensis* (Figure 4D-E).*

Panels D-F also raise more questions. For instance, we ignore the proportion of FT-positive neutrophils in the DHR+ vs DHR- cells. This is a major issue because the proportion of infected cells is changed in WT vs KO mice in figure 1.

*It would be informative to measure DHR and *F. tularensis* infection in the same cell at the same time to correlate infection level with DHR production. However, unfortunately this was not possible due to technical issues (permeabilisation of the cells is required to stain for *F. tularensis* with our APC-conjugated antibody, yet upon permeabilisation intracellular DHR signal is lost). Nevertheless, results from Figure 4B and C show a reduced ROS output in CD200R^{-/-} neutrophils in a situation where cells are not infected with *F. tularensis* (cells are stimulated with PMA). Thus, these results show a clear difference in ROS output between WT and CD200R^{-/-} neutrophils independent of *F. tularensis* infection.*

However, we appreciate the comments regarding experiments measuring ROS output after *F. tularensis* infection. We have repeated and reanalysed the measurement of ROS production in *F. tularensis*-infected neutrophils, using additional markers to better identify viable cells. We find that, in our original experiments, many of the DHR-negative cells were in fact non-viable, and therefore potentially DHR-negative due to leakage of the dye. In the repeat experiments we have gated out non-viable cells (Figure 4D and E), and still show that *CD200R*^{-/-} neutrophils exhibit significantly reduced ROS output during *F. tularensis* infection compared to WT. Thus, in models that are dependent and independent of *F. tularensis* infection, we show that *CD200R*-deficiency results in a reduction in ROS output in neutrophils. We have modified the figure and legend accordingly to incorporate these new data.

Having the DHR123 data for macrophages would also have potentially contributed to better explain the differences in terms of FT burden between both cell types in vivo.

We agree with the reviewer that this experiment would be useful, and have included new data to this end. We show that, in contrast to neutrophils, WT and *CD200R*^{-/-} BMDM infected with *F. tularensis* for 24 hours displayed no significant difference in ROS output (Figure S4A and B). Thus, *CD200R* appears to play different functions in macrophages versus neutrophils. We have added additional text to our manuscript to describe these results (lines 295-297).

Finally the authors claim that altered ROS generation accounts for an increased burden of FT in neutrophils. Yet, definite proof that FT viability is enhanced in neutrophils from *CD200R*-deficient mice is never provided.

We apologise if the wording in our manuscript has led to any confusion. We have shown that during infection, there are a significantly higher % of neutrophils in *CD200R*^{-/-} mice and that a higher % of these neutrophils are infected with *F. tularensis* (Figure 2D and E). We now also add additional data to the manuscript showing that *CD200R*^{-/-} neutrophils display enhanced overall bacterial burden after *F. tularensis* infection in vitro (Figure S1A, new text lines 209-210). Coupled with the fact that depletion of neutrophils in vivo significantly reduces the enhanced lung bacterial burden observed in *CD200R*^{-/-} mice, (Figure 3C), these data strongly suggest that *CD200R*^{-/-} neutrophils are contributing to the enhanced bacterial burden observed in vivo, most likely directly through enhanced infection of these cells.

Our subsequent data highlighting differences in ROS production by *CD200R*^{-/-} neutrophils offer a potential mechanism that could contribute to this enhanced

bacterial burden. However, we cannot conclude definitely that ROS-mediated reduction in viability of bacteria in CD200R^{-/-} neutrophils is the sole reason responsible for enhanced bacterial burden observed. We have added additional text to the discussion of the manuscript to make this clear (lines 349-351).

Regarding supplementary figure 1, a reference should be provided to support that the claimed cell subsets are accurately identified using the proposed gating strategy. Otherwise, cytopins followed by a GEMSA-like staining should be performed to identify the classical populations seen in the lung (alveolar macrophages, vs neutrophils, vs eosinophils & lymphocytes).

We apologise for any lack of clarity here. The gating used has been used successfully by other groups (e.g. Misharin et al. 2013, PMID: 23672262; Yu et al. 2016, PMID: 26938654) and thus we are confident the cell populations have been accurately identified. We have added these references to the manuscript as requested by the reviewer (lines 235-237).

In the discussion the authors write that their results support that CD200R impacts on myeloid cell subset migration, after which they state that absolute numbers of inflammatory cells are increased to deal with FT infection in the lung. If the data regarding absolute numbers of cells is available, it would certainly benefit the validity of the manuscript, especially in the context where the authors intend to make a point that supernumerary neutrophils act as an additional reservoir for FT in the KO mice; which is difficult to support using frequencies, as it is currently the case.

*We agree with the reviewer that data on total cell numbers would strengthen our discussion, and we include new data in the revised manuscript to this end. Thus, we demonstrate that at day 7 p.i. with *F. tularensis* there were significantly increased total neutrophil numbers in the lungs of CD200R^{-/-} mice compared to WT (Figure S3A). Similarly, we saw significantly more *F. tularensis*-infected neutrophils in CD200R^{-/-} mice compared to WT (Figure S3B). In contrast, there was no significant difference in the total numbers and total infected CD11b+ macrophages when between WT and CD200R^{-/-} mice (Figure S3C-D). These data support the findings of increased neutrophil frequencies at day 7 p.i. presented in Figure 2D and Figure 3B. These new data have been added to the manuscript to highlight these results.*

In addition, the notion that CD200R impacts on migration should be better substantiated. How can it be differentiated from recruitment in the context of your study?

We apologise for the lack of clarity in the wording of this in the manuscript as we cannot differentiate between migration and recruitment. We have altered the text in the resubmitted manuscript to better reflect this point (line 357).

CD200R ligands exist for in vitro and in vivo use. This type of approach would have served the concluding sentence of the manuscript much better. In the current form this concluding remark remains highly speculative.

*We thank the reviewer for this important suggestion and (as mentioned previously) have now added new data to better support this claim. Thus, we have treated mice with CD200-Fc recombinant protein is a CD200R agonist (Gorczyński et al 1999, PMID: 10415071), which has been shown to have effects in vivo during other infections such as influenza (Snelgrove et al. 2008, PMID: 18660812). We demonstrate that a single intranasal treatment of CD200-Fc of wild type C57BL/6 mice at day 3 post-infection, coinciding with onset of weight loss, significantly reduced the bacterial burden in the lung at day 7 p.i. with *F. tularensis* compared to control human IgG-treated mice (Figure 2H and I). Furthermore, we see a significant decrease in the percentage of neutrophils following CD200-Fc treatment (Figure 2J). Additional text has been added to the manuscript to describe these results (line 249-258).*

*Thus, having initially shown CD200R^{-/-} mice display an enhanced *F. tularensis* lung burden compared to WT at day 7 p.i. (Figure 2B), these data now show that CD200-Fc treatment of WT mice can decrease *F. tularensis* lung burden. Importantly, this reduced burden is seen when treatment is started 3 days after the initial infection, better reflecting when treatments for infection would be administered. Although further studies are required to optimise the dose and timing of treatment to maximise the beneficial effects, our data nevertheless strongly suggest that the CD200R pathway is a potential therapeutic target for treatment of *F. tularensis* infection.*

Statistics should also state how the assumption of normality and equal variance between groups were assessed prior to choosing parametric tests. This is critical to provide the readership with the insurance that the statistical differences seen at figure 2 A are accurate.

We apologise for the lack of clarity. For all data, D'Agostino-Pearson and Shapiro-Wilk normality tests were conducted before choosing the appropriate statistical test, with parametric tests used only upon data passing normality tests. These details have now been added to the Materials & Methods section (lines 194-195).

Additional comments:

Although ROS contribute to the neutralization of FT, this pathogen is considered to be somehow resistant to the respiratory burst. In spite of the PMA results, it remains difficult to reconcile whether the altered Dihydrorhodamine123 signal results from an altered function of CD200R1-deficient cells, or from their increased FT burden. This should be addressed before concluding in an intrinsic defect in ROS productions in CD200R1-deficient cells upon infection. For instance, is there a loss of difference between the groups in the presence of ROS inhibitors?

*The reviewer is correct that, as neutrophils lacking CD200R expression show greater infection with *F. tularensis*, this is a potential confounding factor in some experiments analysing ROS production. However, our results using control CD200R-sufficient cells (stimulated with PMA) is independent of levels of infection in the cells, and shows a clear reduction in ROS output by CD200R^{-/-} neutrophils (Figure 4B and C). Thus, these results suggest that differences in ROS output by CD200R-sufficient and -deficient neutrophils are evident independent of levels of infection.*

*The use of ROS inhibitors as suggested by the reviewer is a good one. Unfortunately we have had technical issues with determining bacterial burden in such experiments, with ROS inhibitors showing direct anti-microbial activity against *F. tularensis*. We were therefore unable to determine the effects of the inhibitors on *F. tularensis* burden in cells.*

Reviewer 2

The authors should better describe the impact of CD200R deletion on the rate of progression and the severity of infection in their murine model. Does the increased multiplication of *F. tularensis* in neutrophils really affect host resistance to infection?

*We thank the reviewer for this important comment and have added additional data to the manuscript to address this. Thus, CD200R^{-/-} mice show significantly reduced survival at day 7 p.i. compared to WT mice following *F. tularensis* infection (Figure 2A), and displayed significant enhanced spleen size compared to WT mice at this time point (Figure 2C). These data fit with the bacterial burden data showing significant increases in burden later (day 7) but not earlier during infection. Thus, it appears that there is a significant*

effect on host resistance to infection in the CD200R^{-/-} mice, but this effect only manifests later during infection. Additional text has been added to the manuscript describing these new data (lines 222-223 and 226-227).

They should also further describe the potential clinical relevance of their findings in patients suffering from severe pneumonic tularemia. Are there known deficiencies of the CD200 / CD200R pathway in humans? How can one really consider modulating the CD200/CD200R response to improve the prevention of these serious and often rapid infections?

The reviewer raises an important point. There is little known about the deficiencies in CD200/CD200R directly related to human disease and susceptibility to infection, although the pathway has been suggested to be an important marker in a number of human disorders. For example, decreased CD200 and CD200R protein and mRNA expression has been demonstrated in Alzheimer's disease brain tissue (Walker et al. 2009, PMID: 18938162) and high CD200 expression has been suggested as a marker of atypical chronic lymphocytic leukaemia (Falay et al. 2018, PMID: 28713070). Additionally, low CD200 expression on mantle cell lymphoma patients identifies a subgroup with characteristic pathological features (Hu et al. 2018, PMID: 28984300). Thus, although more work is required to determine if there are any human diseases resulting in individuals lacking CD200/CD200R, this pathway does appear useful as a biomarker in some disorders.

*As for the potential for modulating the CD200/CD200R pathway to treat a rapid infection such as tularaemia, we have now performed experiments to directly test whether modulation shows any therapeutic efficacy. We now show that treatment with the CD200R ligand CD200-Fc 3 days after infection results in a significantly reduced bacterial burden and neutrophils in the lung 7 days post-infection (Figure 2H-J). Our treatment at day 3 post-infection is therapeutically relevant, as this would equate to the 3-5 day incubation period of tularemia (Feldman 2003, PMID: 12675294). Thus, although more work is required to find optimal dosing and timing, our results strongly suggest that targeting the CD200-CD200R pathway may be beneficial during *F. tularensis* infection. Additional text has been added to the manuscript describing these results (lines 249-258).*

- Line 60. "however infection via the respiratory route is most virulent,". The bacterium may not be more virulent when infection occurs via the respiratory route. Pneumonic tularemia caused by type A strains of *F. tularensis* is usually characterized by a rapid onset and evolution to a life-threatening infection. However, several reasons may explain higher severity of such infections: higher bacterial inoculum (compared to skin

inoculation via tick bites for example); less effective immune control; etc.

Thanks to the reviewer for this useful suggestion, this has now been re-worded in the text (lines 37-38).

- Line 108. The *F. tularensis* LVS strain is a type B type (subsp. holarctica) strain of *F. tularensis*. Its virulence is highly attenuated in humans, although it remains highly virulent in mice. Moreover, in humans, severe pulmonary infections are usually caused by type A (subsp. tularensis) strains of *F. tularensis*, while type B strains usually cause subacute or chronic pneumonia. Therefore, the LVS strain was not the most appropriate model for the present study. The authors should discuss this limitation in the appropriate section.

The reviewer is correct that using a strain that is highly virulent in humans would be a better model for human disease. Unfortunately, using a highly virulent strain in our study was not possible due to not having the appropriate containment facilities available at our University. We have added additional text stating that future work with a more virulent strain in humans would be beneficial to further the findings of the current study (lines 367-369).

- Line 136. “cells were lysed for 2-3 mins with cold H₂O”. This is not the most common cell lysis methodology used before CFU enumeration. Did authors check that all phagocytic cells were lysed by this procedure? Please specify which temperature was used. A reference should be added.

We used water at 4°C to lyse cells and have added this to our methods section (line 113). This method has been used successfully by others previously as a cell lysis method prior to CFU enumeration (D’Elia et al. 2011, PMID: 21569124; Law et al. 2014, PMID: 25115488; Whelan et al. 2018, PMID: 30296254). To directly address the efficiency of this method in our hands, we enumerated cell lysis in bone marrow-derived macrophages from both WT and CD200R^{-/-} mice. We show that both are lysed to an equal extent, by 4°C H₂O, with greater than 90% lysis achieved (Figure 4 for reviewers). Thus, the method appears to efficiently lyse cells for enumeration of bacterial burden.

Figure 4 for reviewers: Significant cell lysis of BMDM following incubation with 4°C water. WT and CD200R^{-/-}-derived BMDM were incubated with 4°C water for 5 minutes. Cells were counted to determine lysed cells and displayed as percentage lysis. Data represents one experiment (n=5) and is shown as mean ± SD. Statistical analysis was performed using multiple unpaired t-tests (****p<0.0001).

- Line 140. “Female C57BL/6 mice (Charles River, UK) and CD200R^{-/-} mice, developed on a C57BL/6 background”. Are these two models strictly comparable except for CD200R deletion?

The CD200R^{-/-} mice were constructed by Boudakov et al. (2007, PMID: 17667818) on a C57BL/6 background. As we used control C57BL/6 mice from Charles River, we could not directly rule out that differences between control and KO mice were due to some genetic drift of the KO line. However, our new data strongly argues against this possibility. Thus, our studies mentioned above using CD200-Fc as a therapeutic treatment were performed in WT C57BL/6 from Charles River, with results being complimentary to those obtained in CD200R^{-/-} studies (enhanced burden in absence of pathway (Figure 2B), reduced burden when the pathway is activated (Figure 2H-I)). It is therefore very unlikely that any differences obtained between KO and control animals were due to any drift in genetic background.

- Line 143. “8-10 week old mice were infected intranasally (IN) with 50 µl PBS containing ~1000 CFU *F. tularensis*.” Why such a high bacterial inoculum as used for these experiments, while only 10-100 CFU are usually needed to kill all infected mice?

We chose to use a high bacterial inoculum to better understand immunity to F. tularensis that could potentially occur following a severe infection. Others have shown intranasal LVS LD50 to vary from 700 CFU (Fortier et al. 1991, PMID: 1879918; Jia et al 2010, PMID: 20643859), increasing to as high as 3000 CFU in C57BL/6 mice (Ketavarapu et al. 2008, PMID: 18591675). Therefore we chose 1000 CFU as a suitable bacterial inoculum for our model, which worked well in our hands as mice showed progressive worsening signs of infection up until 7 days post-infection in control mice.

- Line 273. “Bacterial burden in the lung showed no significant differences at early time points p.i. (day 1, 3 or 5 p.i.) (Fig. 2A). However, we saw a significantly enhanced bacterial burden at day 7 p.i. in CD200R^{-/-} mice (Fig. 2A).” The authors should further discuss the significance of this finding. How can they explain such a late effect of CD200R defect on F. tularensis infection in mice? Is the observed difference clinically relevant considering that acute pneumonia in humans may be lethal within 3-5 days after disease onset?

The reviewer raises an interesting point. We are unsure why we did not see any early differences in CD200R^{-/-} mice during F. tularensis infection. One possibility is that other cell types (for example macrophages) may play a role earlier during infection and/or other regulatory pathways. Multiple negative regulators, such as TREM2 and SIRP α , are expressed on alveolar macrophages and thus a CD200R defect may have limited influence on these cells early during infection. A different role for CD200R on neutrophils, as well as reduced expression of other regulatory markers, compared to macrophages may reduce the amount of redundancy and thus amplify the influence of a CD200R defect later in infection.

Although the timing of this model is slightly delayed compared to human disease, we have used F. tularensis LVS as a model for the more virulent type A strains, in the same way it is a model for more acute pneumonia seen in humans. Our results showing reduced bacterial burdens in the lung following a single dose of CD200-Fc at day 3 p.i., prior to the onset of disease symptoms, demonstrate a real-world therapeutic application of this pathway.

As mentioned above, we have added additional text to the discussion to point out that future work with type A strains will be beneficial (lines 367-369).

- Line 280. “Interestingly, we found that the increased bacterial burden in CD200R^{-/-} mice was restricted to neutrophils, alongside a dramatic influx of neutrophils into the infected CD200R^{-/-} lung at days 5 and 7”. How can authors explain that bacterial burden was not altered in macrophages, although these cells also express CD200R?

As discussed previously, increased expression of other regulatory markers could result in some redundancy on macrophages. Additionally, we now provide new data showing that there is no significant difference in the level of ROS output in WT and CD200R^{-/-} macrophages following F. tularensis infection (Figure S4A and B), which could potentially also explain why we have not seen any differences in bacterial burden in macrophages. As we saw a reduced ROS output in CD200R^{-/-} neutrophils following F. tularensis infection, this could explain the difference in infection levels between macrophages and neutrophils in vivo. Additional text has been added to the manuscript to address this point. (lines 295-297).

More importantly, F. tularensis has been reported to enhance lifespan of infected neutrophils. How could authors differentiate increased neutrophil lifespan from neutrophil influx in infected CD200R^{-/-} mice?

We thank the reviewer for this comment. It is difficult to definitively determine whether differences we have observed are due to changes in lifespan versus changes in neutrophil influx in vivo. When we performed Annexin V viability staining of neutrophils at day 5 and 7 p.i. in vivo, we saw a slight but significant increase in percentage of pre-apoptotic neutrophils and decrease in live neutrophils in CD200R^{-/-} mice compared to WT at day 5 p.i., with this difference no longer present at day 7 p.i. (Figure S3E-F). Thus, enhanced lifespan of neutrophils does not appear to be contributing to the enhanced infected neutrophil numbers observed in CD200R^{-/-} mice. We have therefore added additional text to the manuscript to describe the new data (lines 242-245).

- Line 299 “Depletion of neutrophils in CD200R^{-/-} mice rescued the bacterial burdens so that it was comparable to those in WT mice (Fig. 3C).” Previous experiments have shown that depletion of neutrophils in F. tularensis-infected mice could reduce pathological effects associated with lung infection. Was there any difference between WT and CD200R^{-/-} mice when both were depleted in neutrophils?

The reviewer is correct that a previous study has shown that inhibiting neutrophil migration into the lung, a phenotype of MMP9^{-/-} mice, reduced pathogenic effects of F. tularensis infection (Malik et al. 2007, PMID: 17202364) However, as this phenotype was not directly due to neutrophil depletion, the reduced lung pathology can not be fully attributed to neutrophils. Studies of neutrophil depletion during F. tularensis infection have shown an infection route-dependent response. Neutrophil depletion significantly increases susceptibility to intravenous and intradermal F. tularensis infection, but not intranasal infection (Sjöstedt et al. 1994, PMID:

8005668; Conlan et al. 2002, PMID: 11855943). Although a recent study showed neutrophil depletion at day 3 p.i. rendered mice highly susceptible to intranasal *F. tularensis* (Steiner et al. 2017, PMID: 28373354), in our experiments, neutrophil depletion prior to infection had no impact on the susceptibility to intranasal infection. Furthermore, no differences were observed between infected WT and CD200R^{-/-} mice when neutrophils were depleted. The study by Steiner et al. was discussed in the manuscript and likely highlights the importance of timing of neutrophil depletion towards susceptibility to infection. Depleting neutrophils once they are the primary infected cell type as done by Steiner et al., appears to significantly alter infection outcome compared to depleting neutrophils before *F. tularensis* infection.

- Lines 342-345. The authors should further discuss if higher bacterial burden in neutrophils from CD200R^{-/-} mice were associated with worse pathological findings in infected lung tissues, or higher death rates in the corresponding mice compared to controls? In other words, was *F. tularensis* infection much more severe in CD200R^{-/-} mice?

*This is an important point, and have added additional data to address the comments. We find that CD200R^{-/-} mice show significantly reduced survival at day 7 p.i. compared to WT mice following *F. tularensis* infection (Figure 2A and new text lines 222-223), and also displayed significantly enhanced splenomegaly compared to WT mice at day 7 p.i (Figure 2C and new text lines 226-227). Thus, it appears that CD200R^{-/-} mice suffer significantly more severe effects after infection with *F. tularensis*.*

- Line 393. “Nevertheless, based upon the evidence presented here, manipulating neutrophils through the modulation of the CD200R pathway could potentially be a novel therapeutic target for the treatment of infections caused by *F. tularensis*.” There are examples of other intracellular human pathogens that modulate the CD200/CD200R pathway to increase their virulence (e.g. *Mycobacterium tuberculosis*, *Brucella* sp.). These examples should be discussed. Could *F. tularensis* modulates this cell pathway as well? Also, a number of molecules have been already used to modulate this CD200/CD200R pathway. Could these compounds be tested in the *F. tularensis* model?

The reviewer is correct that other pathogens have been suggested to modulate the CD200/CD200R pathway. We have added additional text to our discussion to discuss this literature (lines 361-366).

In terms of whether modulation of the pathway could be beneficial, we thank the reviewer for this important suggestion and have added additional data

using a CD200R-agonist as a therapeutic treatment. CD200-Fc is a CD200R agonist (Gorczyński et al 1999, PMID: 10415071), which has been shown to be capable of modulating the CD200R pathway during other respiratory infections (Snelgrove et al. 2008, PMID: 18660812).

*We now demonstrate that a single intranasal treatment of CD200-Fc at day 3 p.i., coinciding with onset of weight loss, significantly reduced the bacterial burden in the lung of C57BL/6 mice at day 7 p.i. with *F. tularensis* compared to hlgG control treatment (Figure 2H-I). Furthermore, we see a significant decrease in the percentage of neutrophils following CD200-Fc treatment (Figure 2J).*

*Thus, having initially shown CD200R^{-/-} mice display an enhanced *F. tularensis* lung burden compared to WT mice at day 7 p.i. (Figure 2B), these new data now show that CD200-Fc treatment of WT mice can significantly decrease *F. tularensis* lung burden. Additional text describing these results has been added to the revised manuscript (lines 249-258). Although further studies are required to optimise the dose and timing of treatment to maximise the beneficial effect of CD200-Fc treatment, it nevertheless confirms that the CD200R pathway is a potential therapeutic target during *F. tularensis* infection.*

Reviewers' comments:

Reviewer #1 (Remarks to the Author):

In my view, this manuscript was greatly enhanced and all of my previous questions were addressed with diligence. I only have minor comments.

Title: I suggest to remove 'expression' because have not studied the regulation of CD200R expression, only the impact of its deletion. Also, 'expulsion' was not addressed per se. the titled should be aligned with your key observations that CD200R deletion enhances FT burden and mortality

Regarding figure 1: Unless the authors are willing to include figure 1 for reviewers in the manuscript, they have to acknowledge that there is likely an absence of CD200 in their in vitro system, and that the in vitro observations wt vs ko could in fact result from a prior adaptation of these cells in vivo, or during maturation (in the case of BMDM).

It should also be stressed in the text that the alleviation of bacterial burden in vivo using the cd200 FC does not detail bacterial burden on a per-cell basis. Thus, the most likely mechanism of action is decreased burden because of decreased neutrophil numbers in this case.

The authors should more clearly highlight that their overall findings are in line with the fact that deficient immunoregulation leading to enhanced neutrophil numbers likely contributes to the enhanced pathology (in the ko animals) by increasing immunopathological damages in addition to acting as an additional reservoir.

Regarding statistical analyses, you have not mentioned that homogeneity of variances was confirmed prior to parametric tests. Tests for normality were added, but not for homogeneity of variance. If need be, the transformation required for normalizing variance also requires to be mentioned.

Reviewer #2 (Remarks to the Author):

The authors have addressed all reviewers' questions and suggestions. The manuscript has been much improved and is now ready for publication.

Max Maurin

Response to reviewers, Casulli et al.:

We thank both reviewers for their time in re-reviewing our manuscript and for providing further supportive comments. Below we present our responses to the comments raised, highlighting where new figures and text has been added.

Reviewer 1

We thank the reviewer in concluding our manuscript was greatly enhanced and all of their previous questions were addressed with diligence. Below we address their minor comments:

Title: I suggest to remove 'expression' because have not studied the regulation of CD200R expression, only the impact of its deletion. Also, 'expulsion' was not addressed per se. the titled should be aligned with your key observations that CD200R deletion enhances FT burden and mortality.

We appreciate the comments of the reviewer and have now reworded the title to closer reflect the key observations of the manuscript (line 1-3).

Regarding figure 1: Unless the authors are willing to include figure 1 for reviewers in the manuscript, they have to acknowledge that there is likely an absence of CD200 in their in vitro system, and that the in vitro observations WT vs KO could in fact result from a prior adaptation of these cells in vivo, or during maturation (in the case of BMDM).

In line with the reviewers comment we have added Figure 1 for reviewers into the manuscript (new Figure S2), with additional text to describe these results and their implications (line 216-224).

It should also be stressed in the text that the alleviation of bacterial burden in vivo using the cd200 FC does not detail bacterial burden on a per-cell basis. Thus, the most likely mechanism of action is decreased burden because of decreased neutrophil numbers in this case.

The reviewer is correct that alleviation of bacterial burdens in vivo using CD200-Fc is not measured on a per-cell basis. We have made this clearer in the text and have highlighted the likely influence of decreased neutrophil influx on bacterial burdens (line 269-270).

The authors should more clearly highlight that their overall findings are in line with the fact that deficient immunoregulation leading to enhanced neutrophil numbers likely contributes to the enhanced pathology (in the

ko animals) by increasing immunopathological damages in addition to acting as an additional reservoir.

We have added additional text further highlighting the important contribution that the neutrophil influx in CD200R^{-/-} mice likely plays in contributing to increasing immunopathology, in addition to neutrophils providing an additional reservoir for the bacterium (line 322-324).

Regarding statistical analyses, you have not mentioned that homogeneity of variances was confirmed prior to parametric tests. Tests for normality were added, but not for homogeneity of variance. If need be, the transformation required for normalizing variance also requires to be mentioned.

We have added additional text in Materials and Methods to highlight that homogeneity of variance was tested by Brown-Forsyth and F tests. If homogeneity of variance tests failed, Brown-Forsyth and Welch's correction tests were conducted where appropriate (line 196-198).

Reviewer 2

We thank the reviewer for concluding the manuscript was much improved and is ready for publication.